# The N-Terminal Region of Cucumber Mosaic Virus 2a Protein Is Involved in the Systemic Infection in *Brassica juncea*

**DOI:** 10.3390/plants13071001

**Published:** 2024-03-31

**Authors:** Tae-Seon Park, Dong-Joo Min, Ji-Soo Park, Jin-Sung Hong

**Affiliations:** Interdisciplinary Program in Smart Agriculture, Kangwon National University, Chuncheon 24341, Republic of Korea; cassel@kangwon.ac.kr (T.-S.P.); perdues0@kangwon.ac.kr (D.-J.M.); jisu_park@kangwon.ac.kr (J.-S.P.)

**Keywords:** Korean red mustard, pseudorecombinant, chimera virus, *Raphauns sativus*, *Brassicaceae*

## Abstract

*Brassica juncea* belongs to the *Brassicaceae* family and is used as both an oilseed and vegetable crop. As only a few studies have reported on the cucumber mosaic virus (CMV) in *B. juncea*, we conducted this study to provide a basic understanding of the *B. juncea* and CMV interactions. *B. juncea*-infecting CMV (CMV-Co6) and non-infecting CMV (CMV-Rs1) were used. To identify the determinants of systemic infection in *B. juncea*, we first constructed infectious clones of CMV-Co6 and CMV-Rs1 and used them as pseudo-recombinants. RNA2 of CMV was identified as an important determinant in *B. juncea* because *B. juncea* were systemically infected with RNA2-containing pseudo-recombinants; CMV-Co6, R/6/R, and R/6/6 were systemically infected *B. juncea*. Subsequently, the amino acids of the 2a and 2b proteins were compared, and a chimeric clone was constructed. The chimeric virus R/6Rns/R6cp, containing the C-terminal region of the 2a protein of CMV-Rs1, still infects *B. juncea*. It is the 2a protein that determines the systemic CMV infection in *B. juncea*, suggesting that conserved 160G and 214A may play a role in systemic CMV infection in *B. juncea*.

## 1. Introduction

Cucumber mosaic virus (CMV) is a plant pathogenic virus of the genus *Cucumovirus* belonging to the *Bromoviridae* family, which is distributed worldwide and has the widest host range of over 1300 plant species in 500 genera and 100 families [1,2,3]. Thus, CMV is considered an economically important virus [4]. Recently, in various crops such as tomatoes, peppers, and passion fruits, the occurrence of plant viruses transmitted mechanically was decreasing while the occurrence of plant viruses transmitted by insect vector was increasing [5,6,7,8]. CMV is transmitted in a non-persistent manner by more than 80 species of aphids, and because of these characteristics, CMV damage to crops is increasing [6,7,9]. The genome of CMV is composed of three RNAs enclosed in spherical virions (ca. 30 nm in diameter) [10,11]. The genome of RNA1 (~3.4 kb), RNA2 (~3.0 kb), and RNA3 (~2.2 kb) of CMV encode five open reading frames. The 1a protein encoded by the RNA1 and the 2a protein encoded by the RNA2 of CMV are viral RNA-dependent RNA-polymerase subunits, respectively, and are involved in CMV RNA replication [10]. The 2b protein is encoded in RNA2 and is known to engage in long-distance movement and RNA silencing suppression [12,13,14]. In RNA2, the 2b protein is transcribed and translated by subgenomic RNA4a because it overlaps with the 2a protein [14]. The 3a protein and coat protein (CP) are encoded in RNA3, and CP is expressed from subgenomic RNA4, which is generated from RNA3 [15]. The 3a protein is required for cell-to-cell movement, and CP is involved in both cell-to-cell movement and long-distance movement [15]. CMV is found in a variety of hosts, and various strains exist with different biological characteristics such as host range, virulence, and aphid transmissibility [2,16,17].

The leaf mustard (*Brassica juncea*) belongs to the family *Brassicaceae* and is native to China but is now widely cultivated in India, Europe, Canada, Australia, Korea, and Japan [18]. *B. juncea* has various common names such as brown mustard, Chinese mustard, Indian mustard, and oriental mustard. In Korea, *B. juncea* is eaten raw or used as a main or sub-ingredient for kimchi [19,20]. In China and India, *B. juncea* is used as both an oilseed and vegetable crop. Reported viruses that infect *B. juncea* include turnip mosaic virus (TuMV, genus *Potyvirus*), youcai mosaic virus (YoMV, genus *Tobamovirus*), and CMV [21,22,23,24,25]. Among them, TuMV is a virus that causes great damage to *B. juncea*, and many studies on it have been conducted [22,23,24]. However, there are no known investigations or related studies on damage caused by YoMV or CMV in *B. juncea*.

Many of the specific amino acids and nucleic acids of CMV genes involved in virulence have been discovered [2]. The 3a protein determined systemic infection in soybean, and CP affected systemic infection in squash [26,27]. In radish (*Raphanus sativus*), 2b protein and C-terminal region of the 2a protein had essential functions for systemic movement [3]. CMV infecting *B. juncea* has been reported since the 1940s [25]. However, CMV infecting *B. juncea* is unknown in Korea, and studies on the interaction between *B. juncea* and CMV are unknown worldwide.

In this study, we used the CMV isolates belonging to CMV subgroup I and showed a distinct infection pattern in *B. juncea*: CMV-Co4, CMV-Co6, and CMV-Rs1 to determine the viral factors involved in systemic infection in *B. juncea*. First, to find which genomic RNA(s) determine(s) systemic infection, we investigate the systemic infection of the two CMV isolates and their pseudo-recombinants in *B. juncea*. For this, the pseudo-recombinants were constructed from the in vitro RNA transcripts derived from the cDNA clones of their genomic RNAs in *Nicotiana benthamiana*. In addition, we investigated the genetic mechanisms underlying systemic infection of CMV using chimeric viruses. Our results revealed that the N-terminal region containing the conserved 160G and 214A of 2a protein determine systemic infection in *B. juncea*.

## 2. Results

### 2.1. CMV-Co4 and CMV-Co6 Systemically Infect B. juncea

We examined the pathogenicity of CMV-Co6, CMV-Rs1, CMV-Co4, and CMV-Fny in the ten host plant species: *B. juncea*, *Capsicum annuum*, *Chenopodium quiona*, *Cucumis sativus*, *Cucurbita pepo*, *N. benthamiana*, *N. tabacum* cv. Xanthi nc, *N. rustica*, *R. sativus*, and *Vigna unguiculata*. The four CMV isolates were mechanically inoculated and detected in the upper leaves at 14 days post-inoculation (dpi) by RT-PCR and back-inoculation of *N. tabacum* cv. Xanthi nc. All plant samples were inoculated in triplicate, and there was no difference in pathogenicity between individuals. The pathogenicity of the four CMV isolates in *R. sativus* was divided into two pathotypes (Table 1). To confirm the infectivity of these four viruses in *B. juncea*, eight *B. juncea* plants were mechanically inoculated with these four viruses, respectively. CMV-Co6 and CMV-Co4 induced systemic chlorosis in *B. juncea,* and these viruses were detected by RT-PCR in all plants (Table 1). To check the infectivity of the detected virus, back-inoculation was performed on *N. tabacum* cv. Xanthi nc. Consistent with the RT-PCR results, *B. juncea* inoculated with CMV-Co4 or CMV-Co6 induced systemic mosaic symptoms in *N. tabacum* cv. Xanthi nc. However, CMV-Rs1 and CMV-Fny did not infect any of the *B. juncea* plants (Table 1). CMV-Co6 and CMV-Rs1 were selected for subsequent experiments.

### 2.2. RNA2 of CMV-Co6 Determines Systemic Infection in B. juncea

To identify the RNA genome segment responsible for systemic CMV infection, infectious cDNA clones of CMV-Co6 and CMV-Rs1 and their pseudo-recombinants were constructed by reassortment. RNA transcripts of pseudo-recombinant viruses were generated using an in vitro transcription system and inoculated into *N. benthamiana*. Symptomatic leaves of the inoculated *N. benthamiana* were used as the inoculum for the experiments. Three pseudo-recombinant viruses were named R/R/6, R/6/R, and R/6/6 using reassorted-origin RNA segments and were inoculated onto *B. juncea*. Although none of the recombinants induced symptoms in *B. juncea* or the parent virus, recombinant R/6/R and R/6/6 containing CMV-Co6 RNA2 spread to the upper leaves (Figure 1, Figure 2 and Figure 3). Recombinant R/R/6 containing only RNA3 of CMV-Co6 was not infected in *B. juncea* (Figure 1, Figure 2 and Figure 3). To clarify the systemic infection of *B. juncea* by RNA2, pseudo-recombinant R/4/R containing RNA2 of CMV-Co4 was constructed and inoculated onto *B. juncea*. Similar to R/6/R, R/4/R were systemically spread in *B. juncea* (Figure 1 and Figure 4). These results suggest that RNA2 is involved in systemic CMV infection of *B. juncea*. Simultaneously, CMV-Co6, CMV-Rs1, and their pseudo-recombinant viruses were tested for infectivity in two *R. sativus* cultivars (Seoho-gold and Yeong-dong). Although there were differences in symptoms, systemic infection was confirmed in the two *R. sativus* cultivars when the virus contained RNA2 and RNA3 of CMV-Co6 (Table 2).

Additionally, local infection was investigated in *R. sativus* of pseudo-recombinant. RT-PCR and back-inoculation were performed using the inoculated or upper leaves of *R. sativus* inoculated with CMV-Rs1, R/6/R, and R/R/6. CMV-Rs1 and R/6/R did not infect *R. sativus* at all, but R/R/6 infected only locally (Table 2).

### 2.3. CMV 2a (but Not 2b) Protein Independently Determines Systemic Infection in B. juncea

To investigate the viral factors of CMV involved in *B. juncea* infection, we compared and analyzed the amino acid sequences of two genes, 2a and 2b, encoded by RNA2. At the nucleotide level, the 2a and 2b sequences of CMV-Co6 and CMV-Rs1 showed 97.6% and 97.7% identity with each other, respectively. However, there were only 6 amino acid differences between CMV-Co6 and CMV-Rs1 in 2a protein (Table 3). Among them, four amino acids, 160G, 214A, 805I, and 832L, common to CMV-Co4 and CMV-Co6, were significantly different from CMV-Rs1 (Table 3). The 2b protein sequences of CMV-Co6 and CMV-Rs1 matched perfectly. In a previous study, the systemic infectivity of CMV to *R. sativus* was determined by the 2b protein and the C-terminal region of the 2a protein [3]. Therefore, we constructed an RNA2 infectious clone, 6Rns, in which the overlapping region of the 2a and 2b proteins were replaced by CMV-Rs1 in CMV-Co6 (Figure 5). In 6Rns, an RNA2 infectious clone, the C-terminal region of 2a of CMV-Co6 was substituted with CMV-Rs1 through *Nco*I and *Stu*I, resulting in two amino acids (805th and 832nd) becoming I805V and L832P (Figure 5). At the same time, an RNA3 R6 clone was constructed in which CP, which is a major determinant of systemic CMV infection in *R. sativus* (family *Brassicaceae*), was substituted (Figure 6). The CP sequence of CMV-Co6 and CMV-Rs1 have 93.9% identity at the nucleotide level and 97.2% identity at the amino acid level. The chimeric pseudo-recombinants consisted of R/R/R6cp, R/6/R6cp, and R/6Rns/R6cp, and their infectivity was confirmed in *N. benthamiana* by in vitro transcription. Three chimeric viruses were maintained in *N. tabacum* cv. Xanthi nc were inoculated into *B. juncea* and the two *R. sativus* cultivars, and systemic infection was identified by RT-PCR and back-inoculation at 15 dpi. R/6/R6cp and R/6Rns/R6cp were systemically infected with *B. juncea* (Figure 1, Figure 2 and Figure 3). In contrast, in the two *R. sativus* cultivars, only the case containing RNA2 and CP of CMV-Co6 (R/6/R6cp) was systemically infected, and R/6Rns/R6cp lost its ability to infect systemically (Table 2). These results suggest that two amino acids, 805I and 832 L, located in the C-terminal region of the 2a protein of CMV-Co6, may be involved in the systemic spread of CMV-Co6 in *R. sativus*, but there are differences in the determinants involved in systemic CMV infection in *B. juncea*.

To estimate the differences between these 2a proteins from the protein structure, the 2a proteins of CMV-Co6, CMV-Rs1, chimeric 6Rns, and predicted R6ns as the reciprocal of 6Rns were modeled using SWISS-MODEL. The four 2a proteins matched the template nonstructural protein 4 RNA-directed RNA polymerase. In the predicted three-dimensional model, some changes were observed because of the substitution of the C-terminal region of the 2a protein (Figure 7). However, the N-terminal region of 2a significantly changed its structure (Figure 7).

Additionally, local infection of chimeric viruses in *B. juncea* and *R. sativus* was analyzed by RT-PCR and back-inoculation using inoculated leaves at 14 dpi. CMV-Rs1 and the chimeric virus R/R/R6cp were not detected in inoculated leaves of *B. juncea*. However, the chimeric viruses R/R/R6cp and R/6Rns/R6cp, which contain the CP of CMV-Co6, only locally infected *R. sativus* and R/R/6 (Table 2).

## 3. Discussion

To date, few studies have analyzed the determinants of CMV infectivity in *Brassicaceae* crops [3,25]. CMV-Co4 and CMV-Co6, isolated from the weed *Commelina communis*, can infect various host plants and induce distinct symptoms (Table 1). In addition, CMV-Co4 and CMV-Co6 systemically infected *R. sativus* and *B. juncea* and induced only mild symptoms (Table 1). These results suggest that due to the wide host range of CMV, cross-infection between various crops occurs, and CMV remain undetected in some plants.

The RNA2 of CMV-Co6 is involved in systemic spread in *B. juncea* and acts in an independent manner, despite differences in the genes encoded by RNA1 and RNA3. In the present work, recombinants R/6/6 and R/6/R containing RNA2 of CMV-Co6 moved systemically in *B. juncea*, but these two pseudo-recombinants did not induce any symptoms compared to CMV-Co6 (Figure 2). These results indicate that RNA2 is required for systemic infection, whereas RNA1 is required for symptom induction. R/R/6 containing RNA3 of CMV-Co6 caused local infection in *R. sativus*. In addition, *R. sativus* was locally infected with the chimeric virus R/R/R6, which contains only the CP of CMV-Co6 (Table 2). These results suggest that CMV in *R. sativus* may be systemically infected by 2a after local infection by CP. Host infection by plant pathogenic viruses requires a series of processes such as plant penetration, host recognition, and uncoating [28,29]. The CP of CMV is involved in cell-to-cell and long-distance movement in cowpeas and tobacco, as well as in determining host adaptation to maize [30,31]. Plants protect themselves against viruses through various mechanisms. Resistance at the single-cell level, termed extreme resistance, is a condition in which viral replication does not occur or occurs at essentially undetectable levels in the inoculated cells [32]. The inoculated leaves of *R. sativus* back-inoculation showed that the viruses containing the CP of CMV-Co6 were still biologically active at 14 dpi. Therefore, at least in *R. sativus*, CP of CMV-Co6 is thought to overcome the extreme resistance of *R. sativus*. This was determined by six amino acids (24A, 28S, 179F, 156A, 188Y, and 205V) in the CMV-Co6 CP. The CMV CP has been studied for functional changes caused by single amino acid changes. CMV CP amino acids 129P and 214G determine cell-to-cell movement in squash, and 129P determines local symptom expression in some plant species [26,33]. Another CMV CP amino acid, 148, affects symptom recovery through phosphorylation, and 162nd affects aphid transmission [34,35]. In *R. sativus*, the systemic infection factors of CMV are suggested to be 17P and 129P of CP [3]. However, in our results, regardless of infection in *R. sativus*, the CPs of CMV-Co6 and CMV-Rs1 were all proline at amino acids 17 and 129. These results suggest that the function of the CMV CP in *R. sativus* may be the result of another interaction beyond a single amino acid level. Similar to the 2a protein, changes in CP were specific to *R. sativus*, highlighting the specific interaction of the plant species with the virus. In addition, previous studies on CMV infectivity in *R. sativus* have used CMV-Y and CMV-D8 [36]. Because CMV-Rs1 is non-infectious, whereas CMV-Y is a locally infecting strain, this study is the first to identify the local determinants of CMV infection in *R. sativus*.

We confirmed that the 2a protein plays a decisive role in the systemic infection of CMV in *B. juncea* using a chimeric infectious clone (Figure 1). It is well known that mainly 3a and 2b proteins play a role in the systemic infection of host plants by CMV [27,37,38]. Although the 2a protein is mainly responsible for RNA replication, some studies have reported that the N-terminal region or the GDD motif contributes to systemic infection [39,40]. Particularly in *R. sativus*, which belongs to the *Brassicaceae* family, CMV requires overlapping regions 2a, 2b, and CP for systemic infection [3,36]. In our study, unlike CMV-Rs1, CMV-Co6 was systemically infected in *R. sativus* and chimeric R/6/R6cp containing RNA2 and the CP of CMV-Co6 (Table 2). However, the chimeric virus R/6Rns/R6cp, in which the C-terminal region of 2a was substituted, lost its ability to infect *R. sativus* (Table 2). This is consistent with previous results showing that the CMV requires the CP and C-terminal region of 2a for systemic infection in *R. sativus* [3]. At the amino acid level, 805I and 832L of the 2a protein were shown to be essential for systemic infection of CMV in *R. sativus* but differed from the previously targeted 2a protein of CMV-D8 (AB179765). These results are thought to be due to structural differences in proteins and specific matches between viruses and hosts. Simultaneously, changes caused by 805I and 832L in the CMV 2a protein did not affect systemic infection in *B. juncea* (Figure 1). Alignment analysis of the 2b and 2a proteins indicated that the N-terminal region of the 2a protein is essential for the CMV to systemically infect *B. juncea*, while the 2b protein can be excluded. 2a protein interacts with 1a protein to form a replicase complex and is negatively modulated by the phosphorylation of its N-terminal region [41]. When compared with the 2a protein, the sequence of R/4/R infected systemically in *B. juncea*; 160G and 214A of the 2a protein, which are targeted as factors for systemic infection in *B. juncea*, are located at potential phosphorylation sites (125–335). Phosphorylation is a post-translational modification that alters protein function and has been shown to play various roles in several viruses, such as the bamboo mosaic virus, brome mosaic virus, cauliflower mosaic virus, and potato virus A [42,43,44,45,46]. We modeled four 2a proteins, CMV-Co6, CMV-Rs1, chimeric 6Rns, and chimeric R6ns, using a protein prediction model to structurally analyze changes in amino acids. The four 2a proteins were modeled as viral RdRp but were structurally changed significantly by the four N-terminal amino acids. Changes in the two amino acids of the C-terminus of the 2a protein appeared to be key for determining systemic infection in *R. sativus*, although the changes were very small in the prediction model. These results suggest that changes, such as phosphorylation of the N-terminus of RNA replicase, affect the systemic infection of CMV, specifically in *B. juncea*. However, the results caused by the C-terminal changes in the 2a protein in *R. sativus* were not interpreted. Therefore, future research is necessary to create point mutants, analyze actual phosphorylation, and identify host counterparts.

*Brassicaceae* includes major economic crops such as Chinese cabbage, radish, and mustard. The genus *Brassica* comprises 37 species. Many *Brassicaceae* crops are consumed as leafy vegetables and are damaged by viruses [18,20,47,48]. CMV does not cause significant damage to *Brassicaceae* crops but is mostly found as a co-infection with other viruses. The 2b protein of CMV is a viral suppressor of RNA silencing (VSR), which is known to intensify symptoms due to a synergistic effect when co-infected with heterogeneous viruses [49,50]. In *N. benthamiana* and *Arabidopsis thaliana*, CMV induced more severe symptoms when co-infected with TuMV [51,52]. Even in *R. sativus*, CMV has been reported to cause systemic infection through co-infection with TuMV [17,52,53]. In addition, CMV-Co6 used in this study induced symptoms in both *B. juncea* and *R. sativus* following a single infection (Figure 2). Therefore, a study of the interaction between CMV and *Brassicaceae* crops is necessary. In this study, we identified the determinants of systemic infection by CMV infecting *B. juncea*; however, further studies on CMV pathogenicity in the *Brassica* genus including bok choy, cabbage, Chinese cabbage, and turnip are needed.

## 4. Materials and Methods

### 4.1. Plant Materials

All plant materials were grown in a growth chamber at 27 °C and 60% humidity. In the growth chamber, a light-dark schedule of 16 h of light and 8 h of darkness was used (LD 16:8). *B. juncea*, *Capsicum annuum*, *Cucumis sativus*, *Cucurbita pepo*, *R. sativus,* and *Vigna unguiculata* seeds commercially sold in Korea were purchased and used.

### 4.2. Virus Source and Mechanical Inoculation

CMV-Co4 and CMV-Co6 were isolated from *C. communis* in Chuncheon, Korea. The two viruses were isolated by single local isolation from *Chenopodium quinoa* and maintained in *N. tabacum* cv. Xanthi nc. CMV-Rs1 was generated by in vitro transcription based on the infectious clones obtained from CMV-Gn [54].

All plants were ground with 0.01 M phosphate buffer (PB, pH7.2), and the sap was inoculated onto at least three leaves of plants at the 5-leaf stage and dusted with carborundum. In the case of *B. juncea*, *C. annuum*, *C. sativus*, *C. pepo*, *R. sativus,* and *V. unguiculata*, only the cotyledons were inoculated with the viruses during the cotyledon stage. All tested viruses were tested for infectivity by mechanically inoculating eight individuals of both *B. juncea* and *R. sativus* in one experiment, and this was repeated three times. That is, the infectivity of one virus was tested on 24 plants.

To distinguish between local and systemic infection, the inoculated and upper leaves were mechanically inoculated onto *N. tabacum* cv. Xanthi nc at 14 dpi, respectively.

### 4.3. RNA Extraction and RT-PCR

Total RNA was extracted from the two-leaf disc. The leaf discs were collected from inoculated leaves or upper leaves of plant materials. Collected leaf discs was homogenized with normally total RNA extraction buffer and isolated with phenol (Sigma, Saint Louis, MO, USA) and Phenol:Chloroform:Isoamyl alcohol (25:24:1) (Bioneer, Daejeon, Republic of Korea), twice [55]. Extracted total RNA was ethanol-precipitated and maintained in a −70 °C deep-freezer. 

All total RNAs were subjected to RT-PCR for virus detection and full-length sequence acquisition, and two-step RT-PCR was carried out. In RT reaction, cDNA was synthesized by M-MLV reverse transcriptase (Promega, Madison, WI, USA) using CP specific 3′ primer at 42 °C for 60 min in the first step and at 94 °C for 5 min in the second step. The cDNA was subjected to PCR using CP specific primer sets at 94 °C for 5 min in the first step and at 34 cycles at 94 °C for 30 s, 50 °C for 30 s, and 72 °C for 1 min in the second step. In the last step at 72 °C for 5 min and in PCR reaction, cDNA was amplified by i-Taq DNA polymerase (iNtRON, Seongnam, Republic of Korea). In the case of RT-PCR for amplification of the full-length sequence, the extension condition was changed to 72 °C for 3 min 30 s in the PCR step. Amplified RT-PCR products were loaded in 1.2% agarose gel stained with MIDORI Green Advance (Nippon genetics, Tokyo, Japan). The primers used in this study are listed in Table 4.

### 4.4. Infectious Clone Construction and In Vitro Transcription

The full-length RNAs of CMV-Co6, CMV-Co4, and CMV-Rs1 were amplified using *Bam*HI-T7 polymerase sequence-tagged 5′ primers and *Pst*I- or *Sph*I-tagged 3′ primers (Table 4). The plasmid vector pUC19 was digested with *Bam*HI-*Pst*I set for RNA1 and RNA3 or *Bam*HI-*Sph*I set for RNA2 and ligated with equally digested RNAs using T4 DNA ligase (Takara bio Korea, Seoul, Republic of Korea). The primers used in this study are listed in Table 4. The ligated DNA products were transformed into *Escherichia coli* dh5α using the heat shock method [56]. The resulting colony was cultured in 4 mL LB media with 8 μL ampicillin (100 μg/mL) for 16 h, and the plasmid was extracted according to the manufacturer’s procedure (Favorgen, Taiwan). All clones constructed were sequenced and used in subsequent experiments. The complete genome sequences were deposited in Genbank with CMV-Rs1 (LC765220, LC765221, and LC765222 accession codes) and CMV-Co6 (LC765223, LC765224, and LC765225). Infectious full-length transcripts were in vitro synthesized using Bacteriophage T7 RNA polymerase (Thermo fisher scientific, Waltham, MA, USA) with Ribo m7G Cap Analog (Promega, Madison, WI, USA), according to the manufacturer’s protocol.

For construction of the RNA2 chimeric infectious clones, the restriction sites *Nco*I (1852) and *Stu*I (2662) were digested in both RNA2 clones of CMV-Co6 and CMV-Rs1. Each fragment of *Nco*I to *Stu*I was cross-inserted into digested infectious clones.

In RNA3, the CP substitution clones were substituted with *Apa*I (1142) and *Pst*I digestion. The digested RNA3 fragment of CMV-Co6 with *Apa*I and *Pst*I was inserted into the digested CMV-Rs1 RNA3 infectious clone with *Apa*I and *Pst*I.

The transcribed RNAs were mixed with PB in a ratio of 1:1:1:3 (RNA1:RNA2:RNA3:PB) and mechanically inoculated on three *N. benthamiana* plants at the 5-leaf stage. At 14 dpi, viral propagation was confirmed by RT-PCR and mechanical back-inoculation on *N. tabacum* cv. Xanthi nc. The RT-PCR product was sequenced (Macrogen, Seoul, Republic of Korea) and showed a perfect sequence match to the infectious clone. Then, *N. tabacum* cv. Xanthi nc, which was confirmed to be infected with the virus, was used as a virus source.

### 4.5. Alignment Analysis

Full-length RNA2 and RNA3 sequences of all CMV isolates were aligned using the MEGA7 tool. The two open reading frames, 2a protein and 2b protein, were translated with the MEGA7 tool and imaged using BioEdit software version 7.2.

### 4.6. 2a Protein Structure Modeling

The 2a protein three-dimensional model was built using SWISS-MODEL Workspace (https://swissmodel.expasy.org/interactive (accessed on 6 April 2023)) [57]. We applied the default setting for all parameters within the algorithms, without any modifications.

## 5. Conclusions

The 2a protein of CMV functions independently for host infection and systemic movement in *B. juncea*, and specific amino acids in the N-terminal region of the 2a protein are involved. Further research is needed to analyze the determinants by which CMV induces symptoms in *B. juncea*.

## Figures and Tables

**Figure 1 plants-13-01001-f001:**
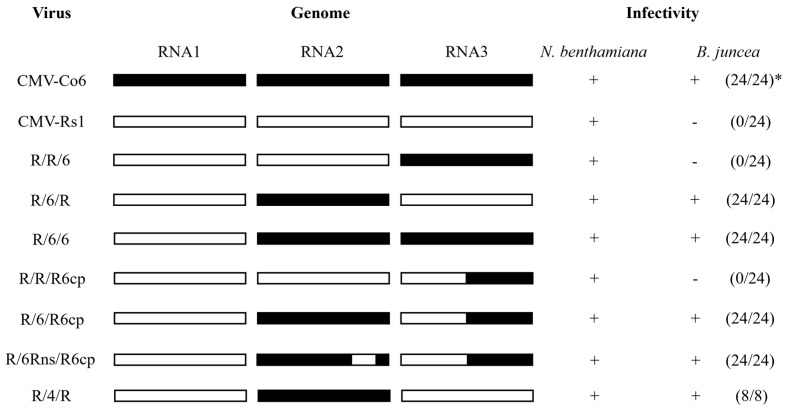
Schematic diagram of used viruses with different RNA and their host response in *N. benthamiana* and *B. juncea*. The three boxes indicate the origin RNA. The black box represents CMV-Co6 or CMV-Co4, and the white box represents CMV-Rs1. Infectivity was confirmed by RT-PCR and back-inoculation. * Number of infected plants/inoculated plants.

**Figure 2 plants-13-01001-f002:**
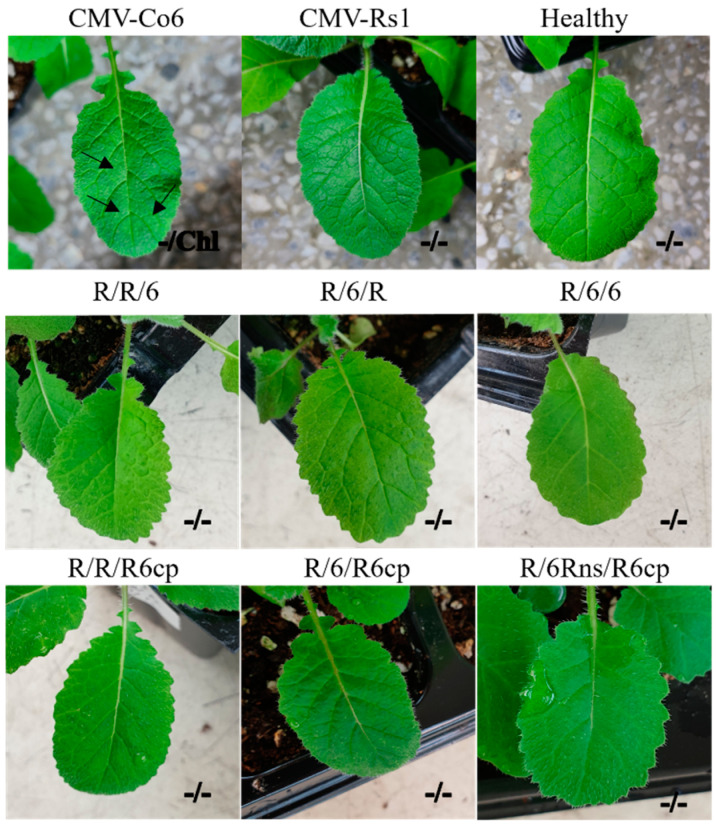
Host response of *B. juncea* inoculated with CMV-Co6, CMV-Rs1, and their pseudo-recombinants and chimeric virus. Photographs were taken 14 days post-inoculation. Only CMV-Co6 induced systemic chlorosis (Chl). − indicates no symptoms. Black arrows indicate the light green island caused by chlorosis.

**Figure 3 plants-13-01001-f003:**
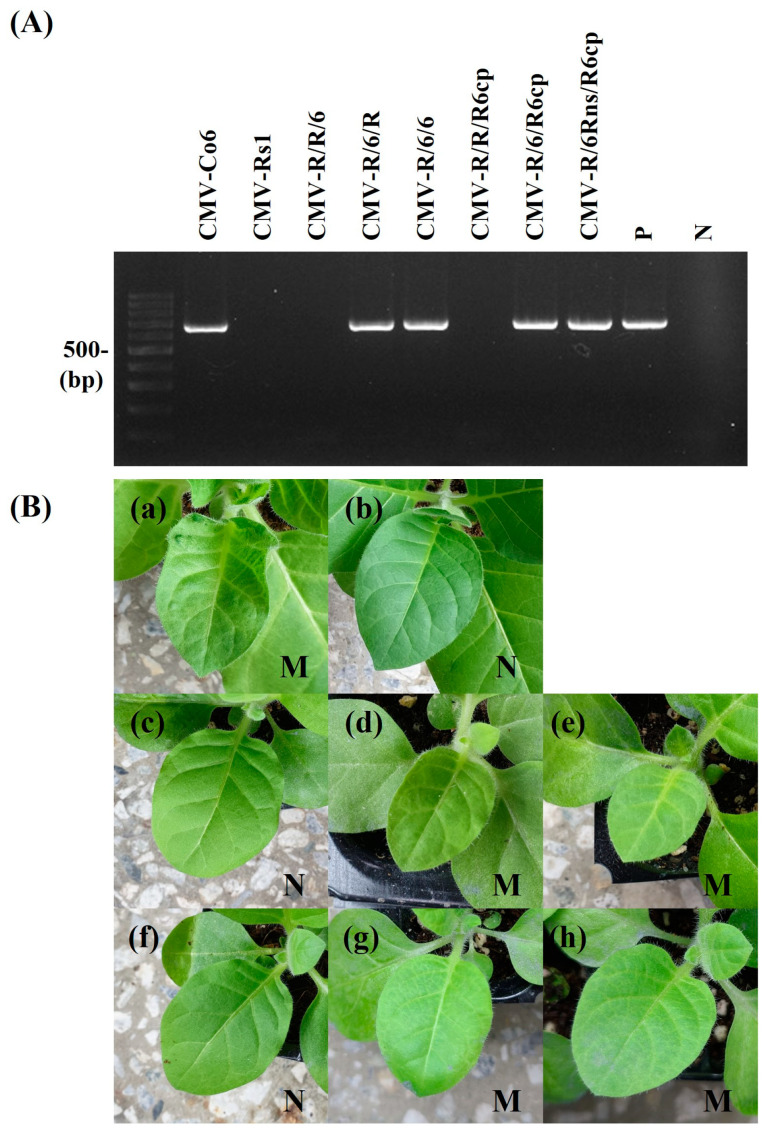
Detection of CMV infectivity in *B. juncea*. CMV-Co6, CMV-Rs1, and their pseudorecombinants and chimeric virus were mechanically inoculated on more than 8 *B. juncea* plants. It was repeated 3 times. At 15 dpi, the upper leaves of inoculated *B. juncea* plants were collected and subjected to RT-PCR (**A**) and back-inoculation on *N. tabacum* cv. Xanthi nc (**B**). (**A**) The resulting RT-PCR products were analyzed by 1.2% agarose gel electrophoresis. Lane M contains the DNA marker, and arrows indicate the expected RT-PCR products (657 bp). Lane P and N indicate the positive control and negative control, respectively. (**B**) The photographs were taken at 7 dpi. M, mosaic in upper leaves; N, no symptom in upper leaves. (**a**), CMV-Co6; (**b**), CMV-Rs1; (**c**), CMV-R/R/6; (**d**), CMV-R/6/R; (**e**), CMV-R/6/6; (**f**), CMV-R/R/R6cp; (**g**), CMV-R/6/R6cp; (**h**), CMV-R/6Rns/R6cp. The pathogenic pattern was consistent with RT-PCR.

**Figure 4 plants-13-01001-f004:**
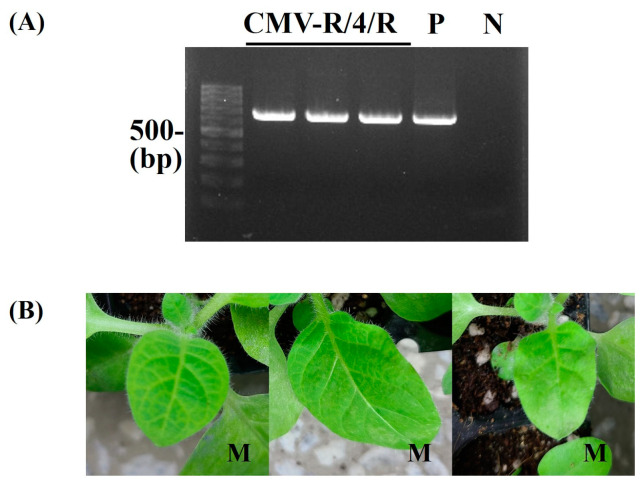
Detection of CMV-R/4/R infectivity in *B. juncea*. CMV-R/4/R was mechanically inoculated on more than two *B. juncea* plants. It was repeated 3 times. At 15 dpi, the upper leaves of inoculated *B. juncea* plants were collected and subjected to RT-PCR (**A**) and back-inoculation on *N. tabacum* cv. Xanthi nc (**B**). (**A**) The resulting RT-PCR products were analyzed by 1.2% agarose gel electrophoresis. Lane M contains the DNA marker, and arrows indicate the expected RT-PCR products (657 bp). Lane P and N indicate the positive control and negative control, respectively. (**B**) The photographs were taken at 5 dpi. M, mosaic in upper leaves. The pathogenic pattern was consistent with RT-PCR.

**Figure 5 plants-13-01001-f005:**
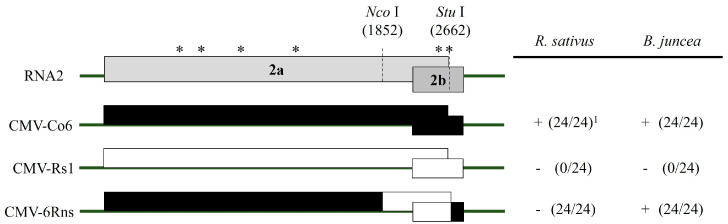
Schematic diagram of genome structure for chimeric RNA2. Structure of the chimeras constructed between CMV-Co6 and CMV-Rs1. Black and white box indicate their origin genome type. The chimeric RNA2 6Rns was generated by exchange of *Nco*I/*Stu*I fragment. Asterisks indicate the six different amino acids. Infectivity was confirmed by RT-PCR and back-inoculation. ^1^ (Number of infected plants/inoculated plants), +, infected; − not infected.

**Figure 6 plants-13-01001-f006:**
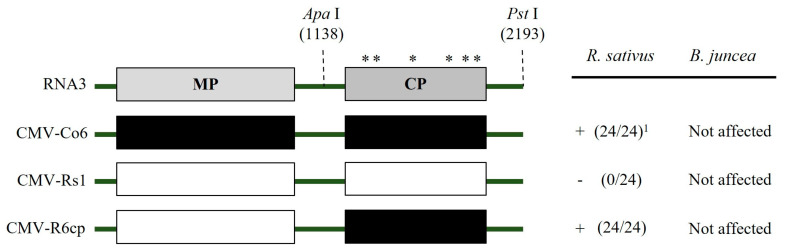
Schematic diagram of genome structure for chimeric RNA3. Structure of the chimeras constructed between CMV-Co6 and CMV-Rs1. The chimeric RNA3 R6cp was generated by exchange of *Apa*I/*Pst*I fragment. Numbers indicate amino acid positions. Asterisks indicate the six different amino acids. ^1^ (Number of infected plants/inoculated plants), +, infected; − not infected.

**Figure 7 plants-13-01001-f007:**
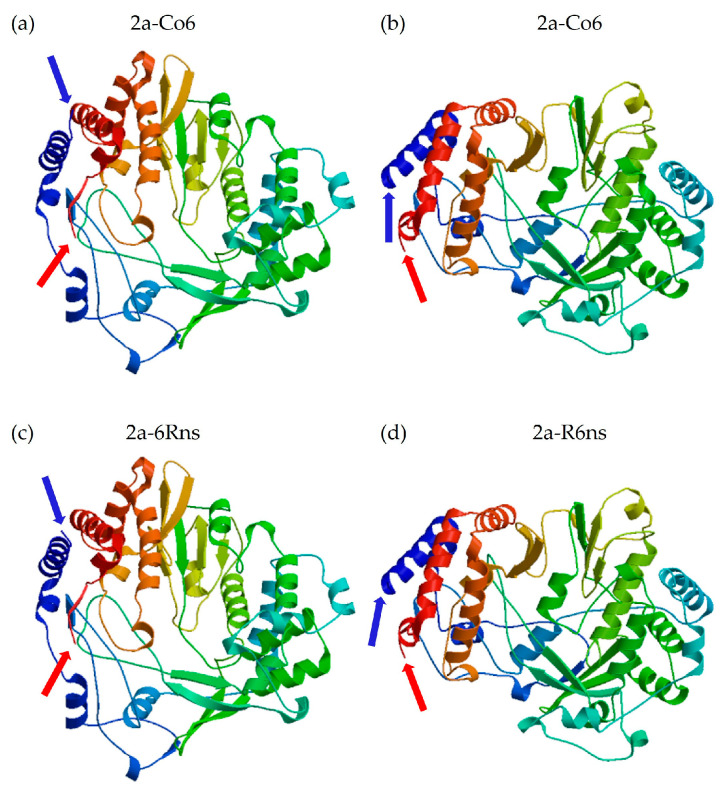
Three-dimensional model of 2a protein. (**a**) CMV-Co6. (**b**) CMV-Rs1. (**c**) Chimeric 6Rns. (**d**) Chimeric R6ns. Blue arrows indicate N-terminal amino acid of 2a protein. Red arrows indicate C-terminal amino acid of 2a protein.

**Table 1 plants-13-01001-t001:** Pathogenicity of four CMV isolates on indicator plants.

Host	Symptoms *			
	CMV-Co6	CMV-Co4	CMV-Fny	CMV-Rs1
*Brassica juncea*	+/Chl	+/Chl	nt/−	−/−
*Capsicum annuum* cv. Cheong-yang	+/M	+/M	+/M	+/M
*Chenopodium quinoa*	L/−	L/−	L/−	L/−
*Cucumis sativus*	+/M	+/M	+/M	+/M
*Cucurbita pepo*	L/N, sM	L/N, sM	+/M	+/M
*Nicotiana benthamiana*	+/M	+/M	+/M	+/M
*N. tabacum* cv. Xanthi nc	+/M	+/M	+/M	+/M
*N. rustica*	+/M	WRS/N, R	+/M	+/M
*Raphanus sativus* cv. Seoho-gold	+/M	+/+	nt/−	−/−
*R. sativus* cv. Yeong-dong	+/M	+/+	nt/−	−/−
*Vigna unguiculata*	L/−	L/−	L/−	L/−

* Inoculated leaves/upper leaves; +, no symptom; − not infected; L, local lesion; M, mosaic; N, necrosis; nt, not tested; R, rugose; sM, severe mosaic; WRS, white ring spot.

**Table 2 plants-13-01001-t002:** Pathogenicity of pseudo-recombinants and chimeric virus of CMV-Co6 and CMV-Rs1 in *B. juncea* and *R. sativus*.

Host	Pathogenicity							
	CMV-Co6	CMV-Rs1	R/R/6	R/6/R	R/6/6	R/R/R6cp	R/6/R6cp	R/6Rns/R6cp
*Brassica juncea*	+/Chl ^a^	−/−	−/−	+/+	+/+	−/−	+/+	+/+
*Raphanus sativus* cv. Seoho-gold	+/M	−/−	+/−	−/−	+/+	+/−	+/+	+/−
*R. sativus* cv. Yeong-dong	+/M	−/−	+/−	−/−	+/+	+/−	+/+	+/−

^a^ Inoculated leaves/upper leaves symptoms. Chl, chlorosis; M, mosaic; +, symptomless confirmed by RT-PCR and back-inoculation; −, no infection confirmed by RT-PCR or back inoculation.

**Table 3 plants-13-01001-t003:** Amino acid differences in 2a protein in RNA2 of CMV-Co6, CMV-Rs1, and CMV-Co4.

Protein	Amino Acid Position	Virus		
		CMV-Co6	CMV-Rs1	CMV-Co4
2a	160 *	Gly	Ser	Gly
	214 *	Ala	Val	Ala
	313	Thr	Ile	Ile
	449	Val	Asp	Asp
	805 *	Ile	Val	Ile
	832 *	Leu	Pro	Leu

Asterisks indicate common amino acid residues of CMV-Co4 and CMV-Co6 that differ from CMV-Rs1.

**Table 4 plants-13-01001-t004:** List of primers used for CMV detection and full genome sequence of CMV isolates.

Primer Name	Nucleotide Sequence (5′ → 3′) *
CMV RNA1 5′ end *Bam*HI T7	CGGGATCC*taatacgactcactata*GTTTTATTTACAAGAGCGTACG
CMV RNA2 5′ end *Bam*HI T7	CGGGATCC*taatacgactcactata*GTTTATTYWCAAGAGCGTA
CMV RNA3 5′ end *Bam*HI T7	CGGGATCC*taatacgactcactata*GTAATCTTACACTGTGTGTGTG
CMV RNA1 and 2 3′ end *Pst*I	GCCTGCAGTGGTCTCCTTTGGAAGCCC
CMV RNA3 3′ end *Sph*I	GCCATGCTGGTCTCCTTTGGAAGCCC
CMV RNA2 2374 5′	AGTTCAGGGTTGAGCGTGT
CMV-CP-5′	ATGGACAAATCTGAATCAACCAG
CMV-CP-3′	TCAGACTGGGAGCACTCCA

* The underlined primer sequences indicate restriction enzyme sites; italicized sequences indicate the T7 promoter.

## Data Availability

Sequence data that support the findings of this study have been deposited in GenBank with the LC765220, LC765221, LC765222, LC765223, LC765224, and LC765225 accession codes.

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
