# Peer review of "The N-Terminal Region of Cucumber Mosaic Virus 2a Protein Is Involved in the Systemic Infection in Brassica juncea"

_plants, 2024, doi:10.3390/plants13071001_

Round 1

Reviewer 1 Report

Comments and Suggestions for Authors

In the submitted manuscript, Park et al., examine the determinants of systemic movement of cucumber mosaic virus in Brassica juncea. This manuscript is similar to a recent paper examing CMV movment in radish, but the authors have expanded on this previous report, and identified potential novel amino acids involved in systemic movement/infection. This manuscript used a variety assays including bioassays, in vitro transcription to generate recombinant viruses, PCR and pathogenicity assays of systemic infection, and protein modelling. Overall the manuscript is well written and concise, and the the conclusions are supported by the results. However, improvements must be made prior to publishing this manuscript

Major issues: 1-It appears a section of the introduction is missing. It begins with the word "thus" implying something came before it. I think the first paragraph might have been deleted. This is a major issue that needs to be addressed.

2- the authors refer to systemic infection, but it isn't clear if this means viral movement or replication. Effort should be made to make this more clear.

3- improved labelling of figures/tables

Minor issues

-First introduction of CMV is not clear. Which virus is this, what family genus?

-Intro should provide more molecular details

-Poor description of genome organization. 1a what? 2b what?

ln 26 - over 90 species of what?

ln 33 ability to infect what? more detail required

ln 34 what does this sentence mean? of course viruses have variation

Table 2 not clear what -/- means

ln 204 systemically

206 hides? reword. suggest remain undetected?

205 "showing" is not appropriate. suggest "due to the"

209 acts in a dependent manner. Dependent to what?

210 capable of replicating in systemic leaves. Is this systemic movement or replication? be more clear

213, but were unable to move and infect systemic leaves/tissues

231-232 not clear? prolines? it is consistent? confusing.

57 N-terminal? not clear.

Figure 6 what do numbers in brackets mean? 

Comments on the Quality of English Language

As mentioned, whole sections of intro are missing. Overall quality is good, but some minor improvements can be made

Reviewer 2 Report

Comments and Suggestions for Authors

This study has merits in establishing the facts that systemic infection is Brassica juncea is governed by N-terminal region of cucumber mosaic virus 2a protein. Through the construction of infectious clones and pseudo-recombinants, the study identify RNA2 of CMV as a key determinant for systemic infection in B. juncea. Further comparative analysis of amino acids of 2a and 2b proteins and designing of a chimeric clone reveal that the 2a protein, principally conserved residues 160G and 214A, plays a substantial role in systemic CMV infection in B. juncea. Clarity could be enhanced by expanding on the significance of the recognized determinants in understanding CMV-B. juncea interactions. Major limitation is lack of effective writing of introduction, review and material and methods.

1.      Introduction is written in a haphazard manner and there is no coherence among the statements and flow of information is erratic.

2.      The details of inoculation dates, timings, number of plants, total replicates, design of experiment is missing from the methodology section.

3.      The virus strain sequence accession numbers should be provided.

4.      How were potential sequence alterations or mutations handled to confirm the reliability and accuracy of the constructed clones?

5.      Please write all the scientific names in appropriate manner

6.      Give latest references and future implications of this study.

7.      Elaborate the methodology part.

8.      Conclusion is missing. 

Comments on the Quality of English Language

Moderate corrections are required. 

Round 2

Reviewer 1 Report

Comments and Suggestions for Authors

In the resubmitted manuscript, Park et al. describe the role of the 2b protein of CMV in systemic movement and replication in brassica species. Full description of improvements and track changes should be listed including line numbers. Author response was not very detailed nor sufficient. While the manuscript has been improved, further improvements to the introduction and writing are required. 

Major issues: Authors should describe the nucleotide and amino acid differences between different strains of CMV. Percent identity required, in addition to the map presented in fig 6 and table 3.

Minor issues

ln 26-28 THis sentence is odd and has unclear significance? Is CMV aphid transmitted?

ln 29 CMV infection is increasing - Can you be more specific? Where? WHat plants?

ln 29/30 are they transmitted independently? poorly written

32 - delete "a process of"

33 incomplete sentence. Should be "encoded on"

37 subgenomic RNA4. This is a bit confusing. Make more of a distinction between RNA4 and RNA4a

63/64 awkward - reword

74, 77 both sentences start with "to confirm infectivity", please reword one.

Comments on the Quality of English Language

Significant improvements can be made

Reviewer 2 Report

Comments and Suggestions for Authors

The authors have revised the manuscript as per the suggested lines and it can be accepted in its current form.

Author Response

Once again, thank you very much for all your comments and suggestions.